# Simultaneous rift-scale inflation of a deep crustal sill network in Afar, East Africa

A. La Rosa [1] ✉, C. Pagli[1], H. Wang [2] ✉, F. Sigmundsson [3], V. Pinel[4] & D. Keir [5,6] ✉

Decades of studies at divergent plate margins have revealed networks of magmatic sills at the crust-mantle boundary. However, a lack of direct observations of deep magma motion limits our understanding of magma inflow from the mantle into the lower crust and the mechanism of sill formation. Here, satellite geodesy reveals rift-scale deformation caused by magma inflow in the deep crust in the Afar rift (East Africa). Simultaneous inflation of four sills, laterally separated by 10s of km and at depths ranging 9–28 km, caused uplift across a ~100-km-wide zone, suggesting the sills are linked to a common mantle source. Our results show the supply of magma into the lower crust is temporally episodic, occurring across a network of sills. This process reflects inherent instability of melt migration through porous mantle flow and may be the fundamental process that builds the thick igneous crust beneath magmatic rifts and rifted margins globally.

Geophysical imaging beneath magmatic rifts and passive margins coupled with geochemical analysis of the associated erupted lavas show that lower crustal intrusion in the form of sill-like bodies is a preferred mechanism of magma accumulation and evolution[1–4]. The process (previously known as underplating) is important since the magmatic addition allows crustal extension to occur with minimal thinning (magma-compensated thinning)[5] while also influencing strain localization and subsidence by making the extending plate denser and warmer[6]. There is wide consensus in modern models of lower crustal intrusion that a series of transient and variably interconnected sills are hosted within a mush made of crystals and partial melt[7,8]. Growing lines of evidence also indicate that such deep magma bodies can feed shallow plumbing systems and dike intrusions during eruptive periods[2,7–9]. However, the processes of magma transport from the mantle to the crust (e.g., resulting in a new pressurization event near the crust–mantle boundary) and the spatial and temporal response of deep-seated interconnected sills to new magma inflows is uncertain as the processes of deep magma motions and the related surface deformation are rarely directly observed[10–15].

In the Afar depression (East Africa), the triple junction of Gulf of Aden (GA), southern Red Sea (RS) and Main Ethiopian Rift (MER) branches is exposed on land (Fig. 1a). Extension in both GA and RS rifts is mainly localized in a series of disconnected, ~20-km-wide, ~50–100-km-long magmatic segments that accommodate up to ~20 mm/y of NE-directed extension by episodic dike intrusion and minor faulting[16–19] (Fig. 1a). The lateral step between the rifts is the largest in the Central Afar rift (CA), with the extension transferred between the Dabbahu–Manda–Harraro (DMH) segment and the Assal-Goubbeth (AG) segment in a 100-km-wide zone of distributed faulting formed from a series of overlapping, normal-fault-dominated, and seismically active grabens[16,20] (Fig. 1a). The crust beneath the CA is ~20–30 km thick, with internal layering and seismic properties (elevated P-wave seismic velocities Vp, and ratios of P- and S-waves seismic velocities, Vp/Vs) consistent with it being continental crust heavily intruded by mafic rock or melt[21,22]. This, along with the crust being around twice as thick as expected from plate stretching models, suggests around half the extension over the ~30 Myr history of rifting has been through magmatic addition to the crust[22].

[1]Dipartimento di Scienze della Terra, Università di Pisa, Pisa 56126, Italy. [2]College of Natural Resources and Environment, South China Agricultural University, Guangzhou, China. [3]Nordic Volcanological Center, Institute of Earth Sciences, University of Iceland, Reykjavik, Iceland. [4]University Grenoble Alpes, University Savoie Mont Blanc, CNRS, IRD, University Gustave Eiffel, ISTerre, Grenoble 38000, France. [5]Dipartimento di Scienze della Terra, Università degli Studi di Firenze, Florence 50121, Italy. [6]School of Ocean and Earth Science, University of Southampton, Southampton, UK. ✉e-mail: alessandro.larosa@dst.unipi.it; ehwang@163.com; D.Keir@soton.ac.uk

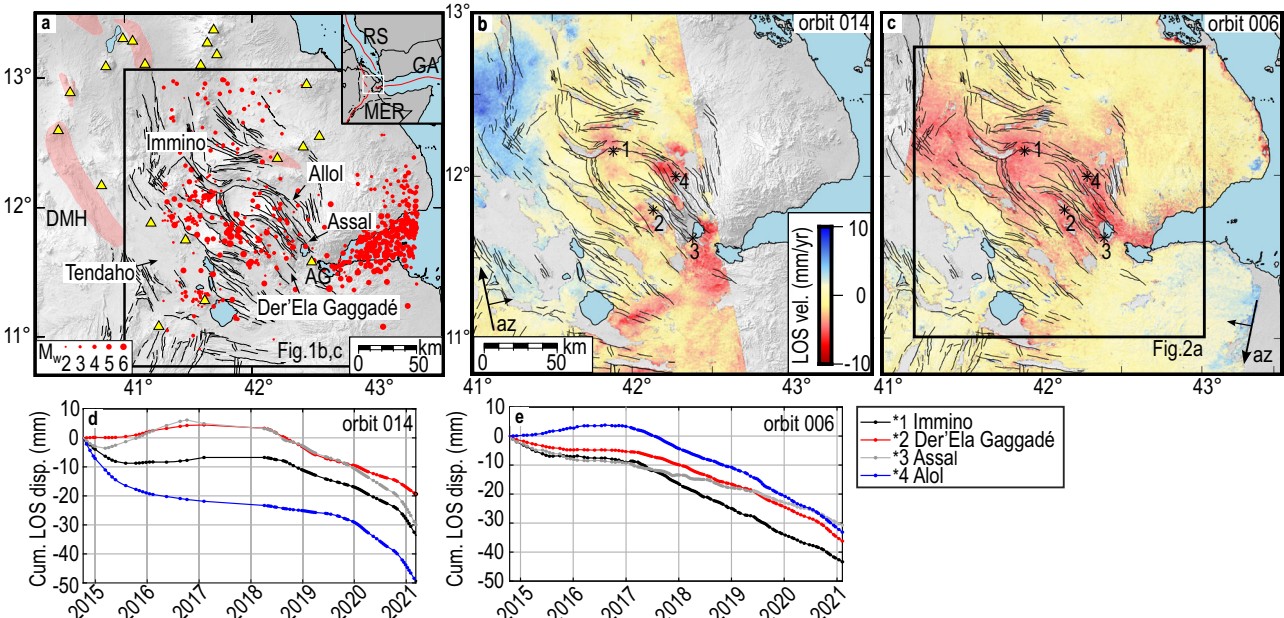

**Fig. 1 | Tectonic setting and InSAR deformation in CA. a** Magmatic segments (red shading), faults (black lines), and Quaternary volcanoes (yellow triangles)[68]. The red dots mark earthquakes in CA from the International Seismological Center (ISC) catalog[69,70] and ref. 34. (Supplementary Data 1). RS Red Sea Rift, GA Gulf of Aden Rift, MER Main Ethiopian Rift, DMH Dabbahu-Manda-Harraro, AG Assal-Goubbeth. **b**, **c** Average LOS velocities (vel.) from ascending 014 and descending 006 orbits, respectively. The black arrows indicate the satellite geometry with the azimuth (az) direction. Black asterisks mark the pixels corresponding to the time series in (**d**) and (**e**). Topography is from the 1 arc-sec (~30 m resolution) Shuttle Radar Topography Mission (SRTM) Digital Elevation Model (DEM)[54]. **d**, **e** Time-series of cumulative LOS displacement (Cum. LOS disp.) from ascending and descending orbits, respectively, for pixels shown in (**b**) and (**c**). Negative values (range decrease) in the LOS data represent ground motions toward the satellite.

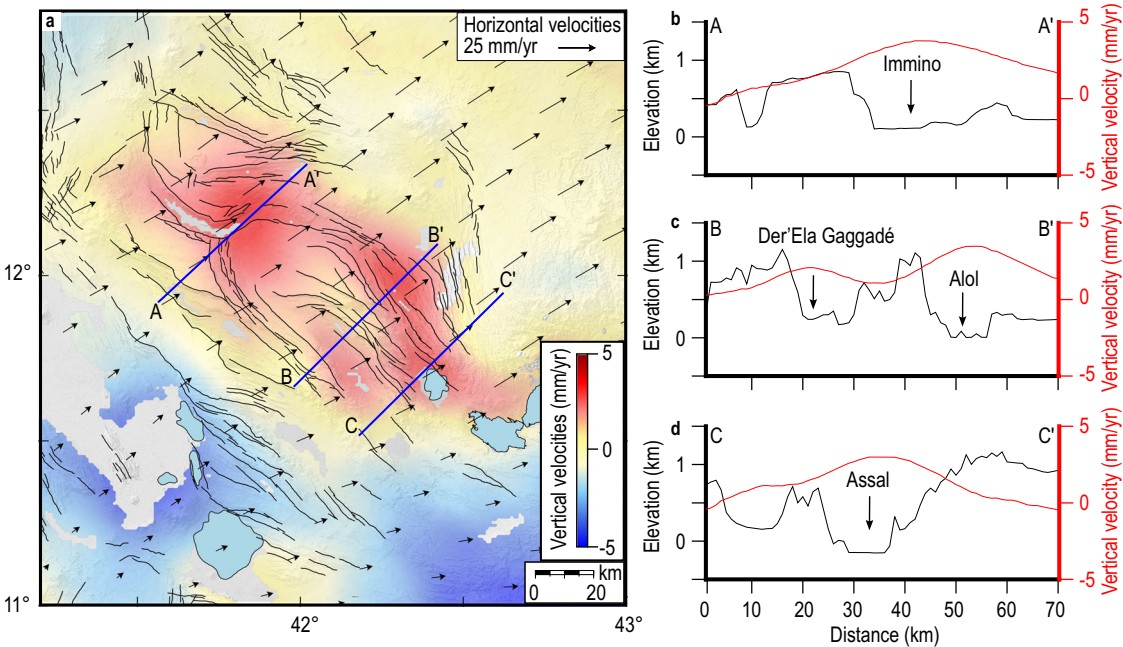

**Fig. 2 | Modeled 3D velocity field of CA. a** Map of vertical surface velocity (positive values for uplift) with horizontal velocities represented as vectors, black lines are major faults. The blue lines mark the location of profiles in (**b**–**d**), showing a comparison between vertical velocities (red) and topography (black) for three key areas in CA. Topography is from the 1 arc-sec SRTM DEM[54].

Interferometric Synthetic Aperture Radar (InSAR) is here used to investigate the deformation caused by deep magma motions beneath CA. InSAR velocity maps were calculated and were combined with the available Global Navigation Satellite System (GNSS) velocities[19,23] to extract the three-dimensional (3D) velocity field in CA with respect to the Nubia plate. We also modeled the InSAR data in different manners: by assuming magma inflation within a network of sills and upward flexure driven by buoyant magma. Additional independent geophysical observations have been used to complement the interpretation of our results.

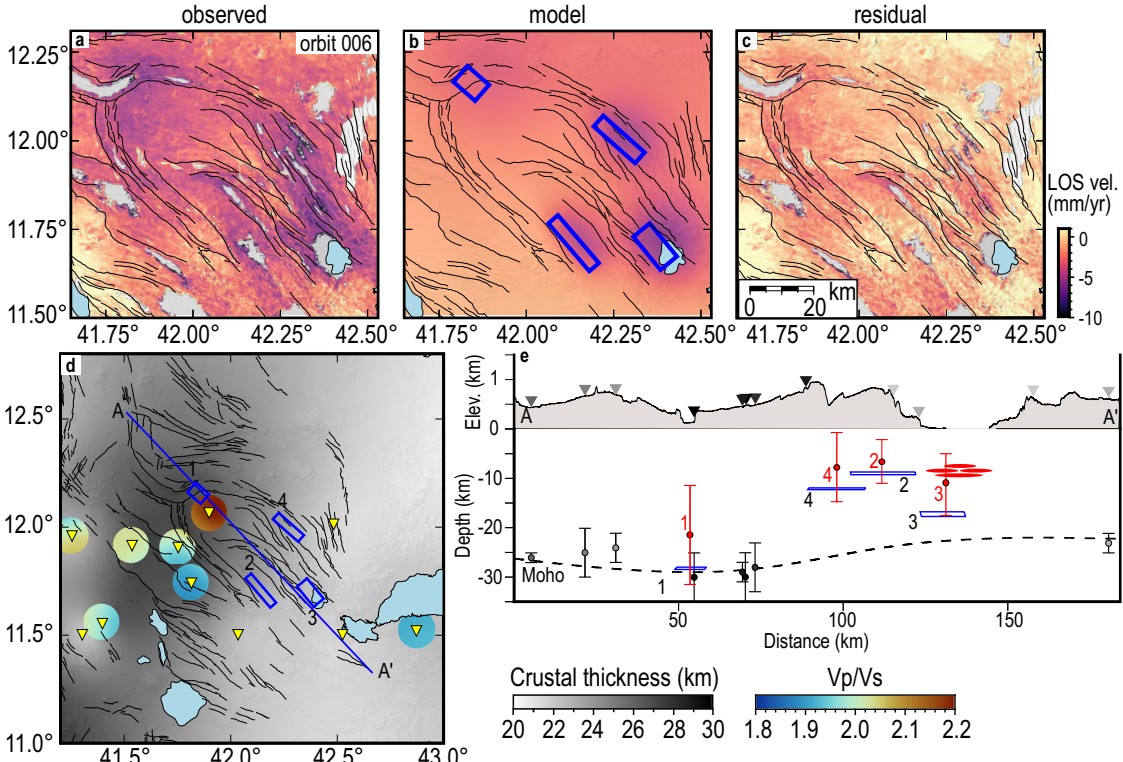

**Fig. 3 | Results of the geodetic modeling.** InSAR observation (**a**), model (**b**), and residual (**c**) for descending orbit 006. The ascending orbit is shown in Supplementary Fig. 12. Negative LOS velocities (vel.) values (range decrease) in the LOS data represent ground motions toward the satellite. **d** comparison between the InSAR model, crustal thickness, and Vp/Vs (when available), as reported by refs. 22,29. Black lines are major faults. The blue polygons in **b** and **d** are the projections of the four best-fit Okada sources (sills) at the surface. The triangles are the stations used by refs. 22,29, while the blue line is the profile track shown in (**e**). Topography is from the 1 arc-sec SRTM DEM[54]. **e** Cross-section showing the comparison between topography elevation (Elev.), sills location (blue polygons), crustal thickness, and related uncertainties when available (black error bars). The red ellipsoids represent the location of the sills at AG inferred by ref. 33 The red error bars are the mean depths (red dot) and standard deviations (2σ, red bars) obtained from the error calculation of Model 2.

## Results

### Deformation in Central Afar

We processed an InSAR dataset made of 255 interferograms from two orbits, ascending and descending, to obtain the time series of cumulative satellite line-of-sight (LOS) deformation, along with maps of average LOS velocities between 2014–2021 (see "Methods" and Supplementary Figs. 1 and 2). We observe the LOS range decrease in four main areas in the CA, with maximum values of ~8 mm/y in both ascending and descending orbits, suggesting uplift (Fig. 1a–c and Supplementary Figs. 2 and 3). Both the raw and filtered time series in these areas (Fig. 1d, e and Supplementary Fig. 2), as also their cross-correlation, show that the uplift was simultaneous between the end of 2016 and the beginning of 2017[15,24] (See "Supplementary methods" and Supplementary Figs. 4 and 5). East of DMH (Fig. 1a–c), the deformation is consistent with accelerated extension following the 2005–2010 rifting episode[18,25,26], and it shows as LOS range increase in ascending (014) and range decrease in descending (006) orbits due to dominant horizontal motions (Fig. 1 and Supplementary Fig. 3).

The InSAR and GNSS velocities were jointly inverted with the aid of a triangular mesh, with a 3 km node spacing, and a Laplacian smoothing factor to obtain the 3D velocity field w.r.t. Nubia (see "Methods", Fig. 2, and Supplementary Figs. 6–11). While the 3D velocity field captures the horizontal motions due to plate-boundary extension[18,19,26] it also showcases four focused uplift patterns within an area about 70 km wide and 150 km long in CA (Fig. 2a and Supplementary Figs. 2, 6, and 7). Horizontal velocities in CA are ~24 and ~13 mm/y in EW and NS components, respectively, as expected for the plate motion of Arabia w.r.t. Nubia[18,19,26]. Vertical velocities reveal uplift with rates of 4–5 mm/y over a ~ 50 km-long, 60 km-wide area covering Immino, Der'Ela Gaggadé, Assal and Alol grabens (Fig. 2b–d). We also inverted pixel-wise ascending and descending LOS velocity maps for the EW and vertical velocities, assuming no NS motions and found similar patterns (Supplementary Fig. 12). The tens of kilometers wide uplift pattern cannot be explained by normal faulting, which would cause subsidence of the hanging-wall and relative uplift of the footwalls. Instead, the graben-wide uplift patterns can be explained by sill inflations[13].

### Geodetic modeling

Surface uplift can be caused by sill inflation within the crust and by upward flexure of an elastic layer driven by buoyant magma at its base[12], and therefore we tested both scenarios. First, we modeled the observations as four inflating sills by jointly inverting the ascending and descending InSAR LOS velocities using a Monte Carlo simulated annealing algorithm, followed by a derivative-based procedure[27]. We assumed four horizontal Okada tensile dislocation models[28] embedded within a uniform elastic half-space with Poisson's ratio of 0.25 and shear modulus of 30 GPa (see "Methods" and Supplementary Figs. 13–15). InSAR velocities at Immino are best fit by a ~9 km × 7 km sill at a depth of ~28 km (sill 1), striking ~N311°E and opening at rates of ~44 mm/y, corresponding to a volume increase of ~2.6 × 10⁶ m³/y (see Fig. 3, Supplementary Fig. 15, and Supplementary Table 1). To the southwest, the modeling finds sills at progressively shallow depths of ~9 and ~17 km at Der'Ela Gaggadé (sill 2) and Alol (sill 3), respectively, and at ~12 km depth at Assal (sill 4) (see Fig. 3, Supplementary Fig. 15, and Supplementary Table 1). These sills are elongated in a NW–SE

direction (~N311°E to ~N320°E), similar to the overall strike of the grabens (Fig. 3). Sills 2 and 3 inflate with rates of ~16 mm/y ($1.2 \times 10^6$ m³/y) and ~42 mm/y ($3.7 \times 10^6$ m³/y), respectively. Inflation rates of ~21 mm/y ($1.6 \times 10^6$ m³/y) characterize sill 4 at Alol (Supplementary Table 1). Our best-fit model fits the observations well as it has a total root mean square (RMS) misfit of ~1 mm/y (Supplementary Table 1). Overall, the depth of the sills follows the trend of the Mohorovičić discontinuity (Moho) in CA with a progressive southeastward shallowing[22,29,30] (Fig. 3). We also explored the non-uniqueness of our best-fit model (Model 2) by calculating the uncertainties (standard deviation, $2\sigma$) on the model parameters with a Monte Carlo simulation of correlated noise[31]. This calculation shows that the model parameters of the four sills have large uncertainties and trade-offs between parameters are also present (Supplementary Figs. 16 and 17). We attribute this to the fact that the uplift signal is relatively small compared to the noise level and that the sills are deep. Nevertheless, the mean depth of the sills from the 100 solutions remains located in the mid-to-lower crust (Fig. 3e and Supplementary Figs. 16 and 17).

We also modeled the InSAR deformation assuming buoyant magma accumulating at the base of the elastic layer and causing flexure (see "Method", and Supplementary Figs. 18 and 19)[12]. We constructed a simplified numerical model of the crust with an elastic layer overlying an inviscid one. We then imposed loading by magma buoyancy at the boundary between the two layers beneath the four areas of observed uplift. We found that such a model always produces a distributed uplift zone of a large spatial extent despite an imposed highly variable topography of the boundary (Supplementary Fig. 15). Such a model is not able to explain the localized and separate uplifts observed in CA. We then tested whether magma buoyancy could explain the residual uplift of the sill inflation model. We found that some buoyancy can explain this long-wavelength residual uplift improving the RMS misfit from 1.35 mm/y to 1.03 mm/y (Supplementary Fig. 18). However, this improvement is minor and within the uncertainties of InSAR. We conclude that the dominant part of the observed uplift is from the sill inflation but that some buoyancy-driven uplift may also contribute.

## Discussion

The InSAR observations and modeling presented here show how a network of magmatic sills inflates in the mid-to-lower crust between the end of December 2016 and the beginning of February 2017, responding simultaneously to an episode of magma inflow from the mantle. At Immino, the inferred location and depth of the sill are in agreement with zones of high (>2.0) Vp/Vs interpreted as magma in the crust[22,29] (Fig. 3d, e). Furthermore, magnetic and gravity observations indicate lower-crustal melt at depths between ~10 km and 28 km beneath the Dobi graben, south of Immino[32]. The inferred depths of sills 2–4 are also in agreement with independent seismic constraints on the depth of crustal intrusions in AG from ref. 33, which inferred the presence of magma at the base of the crust below 8 km depth (Fig. 3). The general trend of the sills follows the southwestward thinning of the crust with the deepest sills 1 and 3 placed near the Moho that has been imaged seismically to range from of ~30 ± 5 km beneath Immino, to ~22 ± 2 km at the active magmatic segment of AG[22,29,30] where sill 3 is located (Fig. 3d). Furthermore, earthquakes from local networks[34–37] show seismicity mainly occurring in a ~10 km thick brittle crust and suggest a brittle–ductile transition to be expected around 10–15 km, where sills 2 and 4 are located (Figs. 1, 3 and Supplementary Fig. 20). Our observations indicate that magma in CA is located at various crustal depths and that the layering near the brittle–ductile transition and the Moho in CA could represent a preferred barrier where magma ponds, evolves and potentially feed magma bodies in the shallower crust[3,4,22,38]. Our InSAR velocities are consistent with previous InSAR measurements in the study area[19,26], and our modeling results have strong similarities with independent geophysical data collected in CA[22,29,30,33,32] and with

modern petrological models of magmatic systems and their natural analogs, where a series of stacked magma bodies forms across the crust, fed by deeper mantle sources[7,8]. Seismic recordings from global catalogs during 2014–2021 show no clear evidence of an increase in the seismic rate after December 2016 (Fig. 1 and Supplementary Fig. 20). This might indicate either that the magma motion was seismically silent or that the low magnitude seismicity typically accompanying intrusions was not recorded by global networks.

Based on the elastic inversion, and assuming the opening rates and geometry of sills in Model 2, we can calculate a total volume change of the four sills of ~0.036 km³ during four years, which is smaller than the ~1.5–2 km³ volume change during the 2005 DMH diking episode, and the ~0.2 km³ volume change in the AG rifting episode in 1978[39,40]. The inferred sills are deep, located either at the depth of the Moho for the deepest (sills 1 and 3) or at the transition between ductile and brittle crust for the shallowest (sills 2 and 4). The lower part of the crust is expected to behave visco-elastically with a relaxation time on the order of 1 to a few years if we consider a viscosity of $10^{18}$ Pas and a rigidity on the order of 10 GPa. For the relatively short time span of our observations (6.5 years), there are no large differences expected between our elastic models of continuously inflating sills and sills located at or near the boundary of elastic and visco-elastic material behavior. The source depth would be similar, but visco-elastic relaxation would reduce the rate of surface uplift for the same rate of magma inflow at depth[41,42]. Therefore, our elastic inversion may lead to an underestimate of the rate of magma inflow into the sills. For such a visco-elastic model, subsidence would be expected after the inflation phase if there is a significant decrease in the rate of inflow at depth. Other possible model configurations may include a sudden pressure increase inside a visco-elastic shell surrounded by an elastic medium[43], consistent with a short pulse of magma inflow, or a sustained increase in pressure in an elastic layer overlying a visco-elastic medium[44], consistent with additional magma inflow into the system throughout the inflation period.

Simultaneous inflation of laterally offset multiple sills beginning in December 2016 suggests that there is a connection between the sills. Such behavior can be well explained by a pressure connection between them and a common source, likely in the mantle. Magma channels/pathways from the common source to the sills may have been in place prior to the onset of inflation. Otherwise, these channels would have formed simultaneously, and magma ascent rates would have been at the fast end of that expected for basaltic melt[45]. The inherent instability of melt migration through porous media has been shown to lead to melt flow focusing in space and time through transient elongated channels in the mantle[46] or porosity/solitary waves, influenced by the relationship between permeability and porosity within the partially molten mantle[47]. Such mechanisms cause episodic melt delivery at the top of the mantle melting column because of processes related to transport, despite melt generation in the mantle occurring at a steady rate[48]. The sills that we modeled are elongated parallel to the rift, similar to shallow sills observed elsewhere in the Afar depression, such as beneath the Erta Ale Ridge[49], or in the Ferrar large igneous province (Antarctica)[50]. In rifts, similar plumbing systems with interconnected sills were also imaged seismically in the Natron rift (East Africa) by refs. 11,14. Pressure gradients caused by episodic magma inflow from the mantle have been recently demonstrated to propagate across interconnected magmatic structures leading to a simultaneous response of sill-like bodies beneath the Hawaiian volcanoes of Kilauea and Mauna Loa in 2019–2021[10], beneath the Askja volcano in Iceland during 2005–2008[51], as also in the Western Galápagos during 2017–2022[15]. In this study we provide geodetic evidence that similar processes can occur at a large scale in continental rift settings.

Until now, investigations of the magma dynamics at continental rifts and at the exposed oceanic ridge of Iceland have been mainly conducted during volcanic eruptions and rifting episodes, showing the

rapid lateral and vertical migrations of magma from deep to shallow reservoirs[2,17,48,51]. These studies also suggested that investigating inter-rifting repose periods might provide further insights into the deeper part of magmatic systems, which still remain poorly resolved[17]. By taking advantage of the lack of shallow eruptive and co-rifting defor-mations, we directly observe the deformation signature from the dynamics of deep magma motions from the upper mantle to the lower crust. We demonstrate that simultaneous, rapid, and widespread arrival of magma occurs in the mid-to-lower crust during non-eruptive periods, responding to pressure pulses from the upper mantle. This mechanism of magma dynamics could be a common means by which magma intrusion compensates crustal thinning in magma-rich rifts to generate anomalously thick crust and may also have a role in the long-term dynamics of rifting episodes and volcanic activity.

## Methods

### InSAR processing
We processed ascending (orbit 014) and descending (orbit 006) interferograms from Sentinel-1a/b acquisitions using the JPL-Caltech InSAR Scientific Computing Environment (ISCE) software package[52]. The dataset covers a time period of ~6.5 years between October 2014 and March 2021. For the ascending and descending orbits, we pro-cessed a total of 104 and 151 interferograms, respectively. We selected interferometric pairs by adopting a Small Baseline Subsets approach that minimizes the spatial and temporal baselines between SAR acquisitions. We also excluded 12-day pairs to avoid phase bias[53] and favored pairs with temporal baselines between 24 days and 144 days, yet longer interferograms (up to 6 months) showing good coherence were kept (Supplementary Fig. 1a and b). Ascending and descending interferograms consist of three frames which were stitched together, and 3 sub-swaths for each frame, covering a maximum area of ~15 × 10⁴ km² and fully overlapping in CA (Supplementary Fig. 1c). For the processing, we coregistered the SLCs and removed the topo-graphic phase using a 1 arc-sec (~30 m resolution) SRTM DEM[54]. We then filtered residual noise and de-correlation using a Goldstein adaptive power spectral filter with a strength of 0.5[55]. Finally, we unwrapped the interferograms using the ICU branch cut algorithm and geocoded them using the 1 arc-sec SRTM DEM. Before the subsequent LOS velocity estimation, each interferogram was visually inspected to identify eventual sudden deforming events (faulting or eruptions) and unwrapping mistakes. When present, the latter were manually fixed.

### Time-series analysis
For each orbit, we reduced the level of noise in the interferograms and produced time-series of incremental and cumulative deformation (Supplementary Data 2–5) along with maps of average LOS velocities and related uncertainties (Supplementary Data 6 and 7) using the Π-RATE software[56]. The interferograms were first cropped to the area of interest, from N9.77°, E40.56° (lower-left corner) to N13.00°, E43.50° (bottom-right corner). Then, to further reduce both phase noise and computing demand, we multi-looked the geocoded interferograms to a pixel size of 90 m. Orbital ramps were removed by fitting them with a linear function estimated following an epoch-by-epoch network approach[57] after masking the active rifts of DMH and CA. A similar network strategy was also used to minimize the topography-correlated atmospheric noise, fitting the linear trend of phase delay with elevation[58]. We applied atmospheric phase screen (APS) filtering using a high-pass Gaussian temporal filter with a cut-off window of 1 year, followed by an adaptive low-pass Butterworth spatial filter with cutoff estimated from a sparse variance–covariance matrix (VCM) of the spatially correlated noise[56] in the masked LOS velocities (Supplemen-tary Fig. 2). We also tested a temporal filter of 0.5 y but results are similar for both ascending and descending orbits (Supplementary Fig. 2). Finally, for each pixel, we calculated the time-series of incre-mental and cumulative deformation along with their uncertainties

using a weighted least-square approach and Laplacian smoothing. In Supplementary Fig. 2, we also provide a comparison between raw and filtered time series. Laplacian smoothing was applied by selecting a smoothing factor that minimizes the trade-off between the solution roughness and the residual sum of squares of deformation. Further-more, for the average LOS velocity maps, we kept only pixels remaining coherent within at least 76 and 104 interferograms and having maximum residual Root-Mean-Square (RMS) misfits of 1 and 0.4 mm/y in orbits 014 and 006, respectively (Supplementary Fig. 3). This allowed us to exclude unstable areas as those covered by deposits. As a final refinement, we visually inspected the average LOS velocity maps and manually masked out small velocity jumps caused by resi-dual unwrapping errors in the original interferograms that were not removed before the inversion.

### Three-dimensional velocity field calculation
We used the VELMAP method[59] to jointly invert the InSAR velocity maps, and available GNSS measurements w.r.t fixed Nubia[19,23] and extract the 3D velocity field and related uncertainties (σ) in CA (Sup-plementary Figs. 6–11). In particular, we used 3D GNSS velocities where available from ref. 16 and 2D measurements elsewhere from ref. 20. Inverting for the 3D velocity field allows for separating the vertical component from the horizontal components of velocity and to better distinguish between tectonic and magmatic deformation. The VELMAP method inverts for the east, north, and vertical velocity components at the nodes of a triangular mesh by solving a system of equations through a weighted least-square approach combined with a Laplacian smoothing operator, as described in ref. 59. Before the inversion, we further multi-looked the LOS velocity maps to a pixel size of 900 m. To better constrain the stable Nubia framework, we also included four fixed GNSS points with velocities equal to zero on the Ethiopian Pla-teau, as also previously done by refs. 18,26. In order to minimize the influence of the co- and post-rifting deformation at DMH, we also removed GNSS measurements covering the time period 2005–2012. We also cropped the ascending orbit 014 along longitude E41.00°, excluding the area of ongoing deformation at DMH[18,25,26] (Supple-mentary Fig. 3). The geodetic datasets were interpolated on a trian-gular mesh with uniform node spacing of 3 km (Supplementary Figs. 6 and 7) and covering the area between N9.40°, E39.45° (lower-left corner) to N13.20°, E43.50° (upper-right corner). We also explored the influence of the triangular mesh design and the GNSS dataset on the 3D velocity field by testing a 5 km spacing (Supplementary Figs. 8 and 9) and by removing all the vertical GNSS measurements in CA (Supplementary Figs. 10 and 11). Both tests do not show significant changes in the 3D velocity field and just small increases of ~1 mm/y in the σ values of each component, indicating that the uplift pattern in CA is a stable feature and it is poorly influenced by the distribution of the 3D GNSS data selected or the mesh design.

### InSAR modeling
We jointly inverted the average LOS velocity maps from both ascending and descending orbits using a Monte Carlo simulated annealing algorithm followed by a derivative-based Quasi-newton approach in order to find the best source parameters that minimize the residual sum of squares between InSAR observations and model[27,31]. Before the inversion, we subsampled the average LOS velocity maps using a quad-tree partitioning algorithm[60] based on a standard devia-tion threshold of 0.35 mm/y for both ascending and descending orbits (Supplementary Fig. 13). The quadtrees were finally refined by manu-ally removing noisy areas that caused higher subsampling. For each test, we conducted exhaustive explorations of the misfit function through 3 standard individual search runs for progressively decreasing annealing temperatures[27]. Modern models of Earth's crust often envi-sage a visco-elastic rheology that better reproduces the lower crustal conditions. In a viscoelastic regime, sudden pressure changes cause an

initial elastic response followed by ductile deformations. While these models perform better in the analysis of long-term deformation processes involving magma motions, it has been demonstrated that the short-term crustal response to magmatic processes can be approximated as elastic[61]. For our purposes, we thus assumed four Okada rectangular tensile dislocation sources[28] within a homogeneous elastic half-space, based on the observation of four maxima on the vertical velocity maps. We weighted the data using the VCM of spatially correlated noise (Model 2) obtained with the time-series analysis for both ascending and descending velocity maps, but unweighted solutions (Model 1) were also tested. We also explored several model solutions by varying the search bounds.

In Model 1, we initially did not apply any weight to the inversion (Supplementary Fig. 14). We explored depth ranges of 2–35 km and set relatively large bounds of the location of the sills' centroids equal to: E41.60–41.90, N11.90–12.33 (sill 1); E42.02–42.22, N11.51–11.82 (sill 2); E42.23–42.48, N11.55–11.83 (sill 3); E41.17–41.34, N11.84–12.10 (sill 4). The bounds on the location of the sills have been chosen in order to be large enough to cover the portions of the patterns with higher deformation but at the same time narrow enough to prevent an overlap between the sill centroids. On the basis of the elongated patterns, strikes were allowed to vary between N270°E and N360°E, letting the sill dip on both sides with maximum angles of 10°. Furthermore, large search bounds of 1–25 km and 1–15 km have been allowed for length and width, respectively. In this solution, the deformation is explained by sills located at progressively shallow depths, from ~28 km at sill 1 to 11–14 km to the southeast at sills 3 and 4. Such sills are tabular and are elongated in NW-SE direction (~N304°E to ~N338°E), similar to the elongation of the grabens (Supplementary Table 1). Model 1 showed a very good fit with the observation with all the parameters, falling within the search bounds, except for the dip angle of sill 4. The model provides low RMS misfits of ~1 mm/y for both ascending and descending orbits (Supplementary Table 1). In Model 2, we used the same bounds as Model 1 and weighted the inversion by introducing the two VCMs of the spatially correlated noise. Overall, the estimated depth, locations, and orientations agree very well with those obtained in Model 1, but in Model 2, all the parameters remain well constrained within the search bounds. Furthermore, the model provides a low RMS of ~1 mm/y for both orbits, as in the previous test. We thus prefer the latter solution. The Model 2 parameters are summarized in Supplementary Table 1 and shown in Supplementary Fig. 15. As a final step, we calculated the uncertainties on the parameters of the four sills using Monte Carlo simulations of the spatially correlated noise[31]. In particular, we generated 100 simulations of the spatially correlated random noise based on the same variance used in the inversion. These simulations were added to the maps of average LOS velocities and inverted[31]. The distribution of parameters for each sill is shown in the Supplementary Figs. 16 and 17.

### Flexure modeling

We tried to reproduce the observed focused surface deformation field by assuming the upward buoyant force induced by magma accumulation at the boundary between an elastic layer and an inviscid layer. The deformation field was determined from the equations for linear elasticity, which we solved using the Finite Element Method in 3D geometry with the COMSOL software. We considered a box $400 \times 400$ km with a mesh refined in correspondence with a boundary topography reproducing the four sills obtained in the elastic Model 2. No deformation is allowed at the lateral boundaries. At the bottom boundary, we applied a Winkler foundation[62] ($\omega = \rho_m g\, U_z$, with the mantle density $\rho_m = 3300$ kg/m$^3$, $g$ the gravitational acceleration, and $U_z$ the vertical displacement of the layer boundary), which corresponds to the buoyant restoring force acting at the bottom of the elastic layer, in a normal direction opposing flexure. For the model,

we used a simplified two-layer setting where the elastic thickness was set to 10 km to reproduce the brittle crust based on the earthquake depth distribution from local networks[34,36,63,64]. We then created four topographic peaks reaching a maximum elevation ($h$) of 24 m in correspondence of the four sills location, to fit the amplitude of surface deformation measured with InSAR. The magma is considered to be stalled under these topographic peaks inducing a vertical upward buoyant force equal to $\Delta\rho \times g \times h \times dS$, with $\Delta\rho$ being the density contrast (50 kg/m$^3$) and $dS$ the unit of surface considered. Such density contrast approximates the values measured by magnetic and gravity surveys between Immino and Tendaho[32]. The 3D deformation was finally reprojected to the satellite LOS from both ascending and descending orbits (Supplementary Fig. 18). While the model fits the amplitude of deformation, the only buoyant force does not reproduce the focused uplift patterns but rather shows a large-scale diffuse pattern. As a final test, we explored the possible combined effect of sill inflation and buoyancy to explain the residual observed in Model 2. To this aim, we rescaled the buoyancy model to fit the amplitude of the long-wavelength residual signals and subtract them from the residual (Supplementary Fig. 19). This test shows that part of the long-wavelength signal could be accounted for by flexure driven by buoyancy, but the majority of focused deformation requires for the sills opening component.

## Data availability

The data generated in this study have been deposited in the OSF repository under the following accession codes (DOI): Supplementary Data 1 and 4 (https://doi.org/10.17605/OSF.IO/6ZM5U)[65]; Supplementary Data 2 (https://doi.org/10.17605/OSF.IO/TUHD6)[66]; Supplementary Data 3–8 (https://doi.org/10.17605/OSF.IO/3FU2R)[67]. These include the earthquake catalog from ref. 34, the InSAR time series, the average LOS velocities, and related uncertainties, the results of the modeling error calculations, and readme files explaining file content and format. The ISC catalog can be downloaded at https://www.isc.ac.uk/iscbulletin/search/catalogue/ using search parameters reported in Supplementary Fig. 20. The Sentinel-1 IW SLCs and satellite orbits files used in this study are provided by the European Space Agency (ESA) and we accessed the files through the Alaska Satellite Facility (ASF) Data Search Vertex (https://search.asf.alaska.edu/#/). The SRTM 30 m DEM used for the figures and the generation of the interferograms can be downloaded from the NASA Earthdata repository at (https://search.earthdata.nasa.gov/search?q=SRTM).

## Code availability

The InSAR Scientific Computing Environment (ISCE) software package v2 is open source and provided by NASA-JPL at https://github.com/isce-framework/isce2. The Π-RATE software (Poly-Interferogram Rate And Time-series Estimator) is open source and available at https://www.phimaging.com/software/pirate/. Figures are produced using Matlab 2022 and the Generic Mapping Tools (GMT) open source software v6.0.0.

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

## Acknowledgements

This study is supported by Ministero dell'Istruzione, dell'Universita e della Ricerca through PRIN Grant 2017P9AT72. C.P. is also supported by the ASI project 'Space it Up!'. A.L.R. and C.P. thank Maurizio Davini and Antonio Cisternino for the IT support. H.W. is supported by the National Natural Science Foundation of China (42274001) and the Associates Program from ICTP/Simons Foundation (284558FY19). D.K. is also supported through NERC grant NE/L013932.

## Author contributions

A.L.R., C.P., and D.K. contributed to the conceptualization of the study. A.L.R., C.P., and H.W. contributed to the InSAR processing and modeling. V.P. and F.S. contributed to the Finite Element Model. All the authors contributed to the investigation and paper writing. A.L.R. and C.P. contributed to the data visualization.

## Competing interests

The authors declare no competing interests.
