## [Peer Review File · Nature Communications]

Simultaneous rift-scale inflation of a deep crustal sill network
in Afar, East AfricaREVIEWER COMMENTS

Reviewer #1 (Remarks to the Author):

ion of a deep crustal sill network in Afar by La Rosa et al., provides an interesting analysis of InSAR and GNSS data showing evidence for the simultaneous inflation of deep sills across central Afar. Combining ascending and descending InSAR observations with GNSS allows the authors to generate a 3D deformation field of central Afar highlighting widespread uplift. These models suggest the simultaneous inflation of 4 discrete sills across the region at varying depths from the Moho to mid crustal levels. The paper is well written and of broad interest and would make a nice addition to nature communications. However, I have a few comments and queries that I feel should be addressed before acceptance of the article.

3D velocity field derivation:

It's not clear from the description whether you are using just the horizontal GNSS in the 3D inversion or both the horizontal and vertical? I note that some sights only have the horizontal components so do you do use a combination of both, i.e. 3D where available and 2D elsewhere? It would be good to a comparison between a version where you only use the horizontal vs one where the vertical are included to get a feel for their influence on the solution.

Similar to the above, did you try doing a simple inversion of just the ascending and descending data to get the EW and vertical? This would also be instructive to see.

Did the authors test any other mesh designs? The final 3D field will be influenced by the mesh used so could be worth exploring.

While it is nice to see the 3D field, why was it not used in the modelling of the sills? The vertical component could be used as a constraint?

InSAR timeseries:

Did you mask any of the deformation prior to doing the corrections on the interferograms with Pirate? The deformation is fairly broad which you may be removing when solving for the linear ramps etc.

While the timeseries in figure 1 are compelling, could you also plot timeseries from other points to demonstrate that the signal is related to just the inferred inflation. There appear to be other zones with apparent movement, but would be nice to see that these aren't showing the same temporal pattern which may invalidate the conclusion that there is a simultaneous inflation.

Also, could you comment on the apparent similarity in timing of uplift? Assuming a common source at the base of the Moho, given the spatial separation of the sills and differences in depth, would you not

expect that there surface expression would be offset? Depending on the flow rate it may be that the InSAR data doesn't have high enough temporal resolution but it may be worth adding something in the discussion.

Modelling:

It's nice to see the approach to test a range of models but I wonder if the truth is actually a combination of the two. Looking at the residuals in Figure 3, there looks to be a systematic underestimate of the vertical. Could there be a combination of buoyant underplating as well as discrete inflation of the sills? You try using the numerical model to remove the longer wavelength signal before the sill estimates or vice versa. This would likely influence the volumes and depths but would be good to test.

Could you also make some comment on the aspect ratio of the sills? Sills 2 and 4 are quite long and skinny, analogue models often show more disc like bodies rather than relatively narrow sills. This may be related to the regional stress field but may be worth having a sentence or two in the discussion.

Figures:

Although the colour scales in Figure 1 and 3 are the same, the LOS displacement shown in Figure 3 looks much bigger than in Figure 1. Could you check the colour bar used?

Reviewer #2 (Remarks to the Author):

These are fascinating observations that I think will be of broad interest to readers of Nature Communications. I think that studying inter-eruption/rifting periods is very important and this work shows that it can provide insights into deeper processes than are otherwise observable. The observations of simultaneous uplift at four deep sills is convincing, but I have queries about a couple details of the analysis of the data, and have also made some line-by-line suggestions for clarifications.

- You point to simultaneous uplift as evidence that the four sills are responding to the same variations in deep melt supply, which I find convincing. However, I wonder whether you have investigated the relationship between the displacement time series above the four sills in more detail, to assess how robust the observations of similarity is, and assess when the time series first become correlated? Is the time of uplift onset actually the same in each location or is there a lag? From Figure 1d and e, I think the situation seems more complicated than described at line 70 as "range decrease in 2014, followed by a stable period and a rapid range decrease from 2017 to 2021". It looks like range decrease is apparent earlier at Alol and Immino in the ascending data? Could you make an assessment of time series similarity using correlation analysis or at least compare the timings of the onset of uplift using time series turning points?

- How have you identified the four specific sill locations for modelling? The methods do not state the

range over which Lat and Lon were allowed to vary – were they then fixed before inversion? More generally, how did you decide that there were sills precisely at sites 1-4? These don't look like the locations of maximum LOS uplift rate From Figure 1, but perhaps maximum vertical velocities from joint inversion with GNSS as show in Figure 2. Is the choice of sill location related to graben axes? Please could this choice be explained in the main text of the paper. The residuals in Figure 3c look systematic, with range increase localised along a few faults and a general underestimation of uplift in the central area – it would be good to include a short discussion of how you interpret that.

General point about figures: I don't find the diverging colour scale very helpful for picking out details of a displacement field that shows only range decrease, as only a small fraction of the colour scale is used (Figures 1 and 3 and several in the supplementary, especially Suppl. 9). Please consider using a different colour scale throughout the paper, and certainly for those figures where range change is only in one direction.

Could you put the uplift (or estimated intrusion rates) in the context of Central Afar's recent geodetic history? For example, how to the rates of volume increase in the four sills compare to intrusive processes during the last episode of rifting?

Line 65. Please specify number of ifgms/time period instead of 'vast'

Figure 3 caption. Please specify what models are shown here – Okada solutions for the four sills only? Please see comments about colour scales.

Line 110, 133-134. "The depth of the sills follows the trend of the Moho in CA with a progressive southeastward shallowing" – This is generally true, though sill 3 is deeper than sills 2 or 4 – can you comment on this?

Line 131. Could you clarify what is meant by network? To me, this implies that the sills are connected to each other directly rather than responding to the same pulse of ascending melt, as described elsewhere.

Line 136. 'Locates' > is located? or is currently accumulating?

Line 144. 'deepest' refers to just sills 3? Can you point to evidence or the depth of the ductile to brittle transition?

Line 168. We have also seen something very similar in the Western Galapagos: <https://www.nature.com/articles/s41467-023-42157-x> I think, if you see fit, that the paragraph at 171-182, could be expanded to address settings beyond rifts, given recent similar observations in Hawai'i and the Galápagos.

Line 385. Please clarify at the beginning of the paragraph that 'Models 1 and 2' refer to weighted and unweighted inversions.

Reviewer #3 (Remarks to the Author):

Reviewer #4 (Remarks to the Author):

Manuscript Title: Simultaneous Rift Scale Inflation of a Deep Crustal Sill Network in Afar

In this paper, the authors use space-based geodetic signals to detect and analyze temporal uplift signals across the rift basins in the Afar Depression, East Africa where continental extension is approaching seafloor spreading. The study area is inarguably unique, and the horizontal and vertical surface elevation changes some of the largest/strongest worldwide. The authors attribute these surface uplifts to episodic inflation of upwelling magma into a network of active sills located at mid to lower crustal depths, and promote dike-sill intrusion models for continental extension. The authors jointly invert InSAR and GNSS data to argue for 4 sills, and to model (1) the location of the sills, (2) the dimensions of the sills, (3) connectivity of sills, and (4) crustal response to inflating and deflating sill complex in varying rheological crustal conditions. Their 3-D surface velocity change models are best fit by simultaneous inflation of a network of active sills at mid-lower crustal depths across a 100 km-wide zone. They infer a common magma source at depths below the Moho feeding them in an episodic manner.

The paper is well written, and the arguments address well-established questions in continental rifting. There remain, however, omissions in terms of model assumptions, cumulative model uncertainties, choice of independent constraints used to validate their models and some input model parameters as noted below. It's clear the authors have thought about these omissions, but we point out that the broad Nature Comms audience will need this information to absorb the results. Although one interpretation seems to be over-stated, but the paper remains strong, and is acceptable with toned down interpretations and some qualifications.

Major concerns

- Our (grad student + advisor review) primary concern is that the authors have not clearly stated that potential field models of temporal changes in surface elevation are non-unique, and that the signal decays rapidly with depth to source body. Could we ever detect a signal from a magma body at depths of 20 km or greater? What volumetric changes would be required to detect an inflation at 40 km?
- Following on from above, why choose as alternative model of an intrusion below the plate, with source depth not stated in the text, and not justified with evidence from exhumed passive margins. Technically speaking, this would be a dynamic source.
- Why 4 sources? Why not 3 or 5? This is key: can the arguments be squeezed into a short paper format?

- In the elastic half space model domain, the authors decided to use Poissons ratio of “0.25”. Considering that bulk crustal V_p/V_s in the Afar Depression is ≥ 1.8 , Poissons ratio of ≥ 0.28 should be used. Poisson ratio of 0.25 corresponds to V_p/V_s of 1.73 which might be an underestimation of their model’s elastic domain.

- In their InSAR and GNSS joint inversions, a constant nodal spacing of 3 km was used in their triangular mesh. The GNSS station spacing in the region is much greater than 3 km as shown in Figure SM3. Do the authors think that the jointly inverted 3-D models have spatial resolution of 3 km?

- 134-136: The comments about the sill depths mirroring Moho depths seems an over-interpretation, given the non-uniqueness of the sill models. If there were independent MT or seismicity data, we could re-consider.

- 144-146; 154-157: Where is the brittle ductile boundary, and given the rapid stressing magmatic zones, would historical seismicity constrain this boundary. I would be more cautious here and focus on key and inarguable points. For example, rapid stressing of visco-elastic material would induce earthquakes, but nowhere do we see evidence of elevated earthquake levels.

- 169-170: The claim that this is the first paper to document inter-actions between magma chambers in a continental rift is a bit over-stated. Gelai and Oldoinyo Lengai volcanoes in the weakly extended N Tanzania rift occurred in 2005-6, and interactions between sill complexes are documented in Oliva et al. GRL, 2019; and Reiss et al., Frontiers, 2021.

Minor/editing Comments

- 24: suggest you replace preferred with common

- 35: Readers won’t understand ‘Afar’: The Afar depression encompasses sub-aerial sectors of the Red Sea and Gulf of Aden rifts.

- 44: mafic or molten or both? How will they influence V_p/V_s and Poisson’s ratio?

- 52: Sills in the lithosphere? What is lithospheric thickness beneath Afar? Why are models indicated as below the lithosphere – confusing.

- Sill intrusions at depth may be seismically silent. This should be stated up front – one of the co-authors is a seismologist.

- Fig 1: Text and graph data are very small.

- Refs 13-16: too much reliance on rates from modelling, and not enough from data. Include key data from Viltres et al., 2020. Viltres, R., Jónsson, S., Ruch, J., Doubre, C., Reilinger, R., Floyd, M., & Ogubazghi, G. (2020). Kinematics and deformation of the southern Red Sea region from GPS observations. *Geophysical Journal International*, 221(3), 2143-2154.

- Since the paper is using geophysical constraints to validate their 3-D models while defining the major physical boundaries in their model domain (e.g., Moho and base of plate), I would suggest the authors to use physical boundaries consistently throughout the manuscript. Instead of just saying “mantle” specify whether it is “mantle lithosphere” or “asthenosphere”. Additionally, while the authors attempt to describe the plate in their model domain in terms of the elastic, brittle-ductile and visco-elastic domains, adding terminologies such as upper, middle, or lower crust and upper mantle will help readers easily follow your arguments.

- 103: How can a model ‘find’: What measure of goodness of fit was use to determine preferred models?

- Supplementary Material: Fig 3: The ‘shaded’ area requires a regional expert to spot.

- We are not InSAR specialists and cannot contribute to details on lines 340-404. But, we note the

reliance on models for rates of rift opening, rather than independent data. GNSS models of Viltres et al. 2020 should be cited.

- Why would the authors consider a melt influx below the plate and into the asthenosphere where viscosity is only crudely constrained? Why don't you just show that a sub-crustal influx is undetectable, and consider this an important result: we can't see deep inflation events beneath rift zones? InSAR can't 'see' to these depths. Isostasy relates to density variations within the plate, not below the plate.
- Supplementary material captions: These need more details and justifications. The SM needs a clear statement about non-uniqueness of models
- SM: Needs comments about seismicity: Are data available and do they inform this work? The 6th author is a seismologist, but it's hard to see his impact on the paper.

Martin Musila and Cynthia Ebinger

We thank the four Reviewers for sending constructive comments to our manuscript “Simultaneous rift scale inflation of a deep crustal sill network in Afar” [Paper NCOMMS-23-46080]. We have addressed all the comments and have incorporated them into the revised version of the manuscript.

In particular, we did additional tests on the 3D velocity field calculation and modeling, both which support our original interpretation of magma intrusion in mid-lower crustal sills. We also clarified the processing strategy and improved the legibility of figures following the Reviewers’ suggestion. We finally included a correlation analysis of our time-series that strengthens our interpretation of simultaneous sill inflation in Central Afar. Some additional figures and tests are included in the following rebuttal while others are incorporated in the manuscript. We think that our analysis provides more convincing evidence and constraints on the presence of simultaneous deep magma intrusions in Central Afar. We are confident that the manuscript has improved as a result of taking on board the reviewers’ suggestions and hope it can now meet the high standards of Nature Communications.

We have responded to the Reviewers’ comments in the following point-by-point rebuttal. Similar comments have been addressed together while minor reviews (e.g., typos, grammar errors) have been directly included in the revised version of the manuscript. The line numbering below refers to the track-changed version of the manuscript.

Reviewer #1

Comment 1: 3D velocity field derivation: *It’s not clear from the description whether you are using just the horizontal GNSS in the 3D inversion or both the horizontal and vertical? I note that some sights only have the horizontal components so do you do use a combination of both, i.e. 3D where available and 2D elsewhere? It would be good to a comparison between a version where you only use the horizontal vs one where the vertical are included to get a feel for their influence on the solution.*

Response1: For the 3D velocity field inversion, we used a combination of 2D and 3D GNSS velocities by Doubre et al. (2017) (2D measurements only) and King et al. (2019) (2D and 3D measurements). We clarified this in the Method section at lines 435-436 by including the following sentence “In particular, we used 3D GNSS velocities where available from ref. 16 and 2D measurements elsewhere from ref. 20.”.

Doing a 3D velocity field inversion without any GNSS vertical velocity as input provides unstable estimates of the vertical component of the 3D velocity field. This is because in the 3D velocity field inversion: 1) we correct residual orbital errors in the InSAR velocity maps with a 3D planar correction using GNSS data as input, hence the vertical component is key for estimating correctly the parameters of the plane, and 2) most of the noise will be mapped onto the vertical velocities if excluding 3D GNSS due to lack of constraints in that component. Therefore, some GNSS vertical velocities are needed for 3D velocity field inversion. However, to address the reviewer comment we excluded all the 3D measurements in Central Afar and left just ten GNSS vertical velocities at the margins of our study area. See new Supplementary Figs. 10 and 11 in which we show the resulting 3D velocity field and related uncertainties (standard deviation, σ). The results show that the uplift pattern in Central Afar remains unchanged compared to our original solution, with just small increases of ~ 1 mm/yr in the σ values of each component (See also new Supplementary Figs. 6 and 7 where we showed the results and σ values of our original solution). This indicates that the uplift pattern in Central Afar is a stable feature and it poorly influenced by the distribution of the 3D GNSS data. We included this test as Supplementary Figs. 10 and 11 and described in the method section at lines 450-455.

Comment 2: *Similar to the above, did you try doing a simple inversion of just the ascending and descending data to get the EW and vertical? This would also be instructive to see.*

Response 2: We inverted the ascending and descending InSAR velocity maps for the EW and vertical components, assuming no NS motions. The vertical velocities are similar in both solutions (Fig. 1 and new Supplementary Fig. 12 in the manuscript), in particular in Central Afar, with the only significant difference being the subsidence inside the Tendaho rift that is not fully retrieved by the simple inversion due to lack of coherent pixels. Instead, it is observed in the 3D inversion (Fig. 1 in the manuscript) since an interpolation basis is used to obtain a continuous velocity field. We stress that we never analyzed the Tendaho signal nor any signal that is not coherent in the original ascending and descending InSAR data.

Regarding the horizontal motions, the two EW velocity maps are consistent and both show the accelerated eastward motion east of Dabbahu. We stress that the EW velocity map from the simple inversion is relative to a fixed point in the given InSAR data while in our 3D inversion we use the GNSS data to define the reference frame and these maps are relative to stable Nubia. As a conclusion, this test supports our original 3D velocity field calculation in Central Afar. We included this test as Supplementary Fig. 12 and described it at lines 87-89.

Comment 3: *Did the authors test any other mesh designs? The final 3D field will be influenced by the mesh used so could be worth exploring.*

Response 3: In the manuscript we use 3 km node spacing, and we have now tested solutions with a larger spacing (5 km) between the nodes, while maintaining a uniform mesh design over the entire area. As can be seen in the new Supplementary Figs. 8 and 9, the larger mesh spacing provides similar but more smoothed patterns of the 3D velocity field and similar standard deviations compared to our original maps that have 3 km mesh spacing (Supplementary Figure 6). Thus, this test indicates that the 3D velocity patterns are not significantly affected by changing the spacing of the mesh. We stress that our 3D inversions are done using a uniform mesh. Irregular meshes, with nodes denser near the center of the study area and sparser in the distal areas are sometimes needed for computational reasons and are known to influence the 3D velocity field. These designs have not been used in our inversions. We now included this inversion test and related uncertainties as Supplementary Figs. 8 and 9 and described it at in the method section at lines 449-455.

Comment 4: *While it is nice to see the 3D field, why was it not used in the modelling of the sills? The vertical component could be used as a constraint?*

Response 4: We decided to invert the LOS velocity maps because we consider it the safer approach. The continuous 3D field is better to represent regional deformation features, while the LOS velocity maps have higher spatial resolution (90 m) that better captures the variability of deformation signals at the scale of the sills. In addition, the vertical components of the 3D field are inverted by ascending and descending LOS rate maps, which results in the joint use of LOS providing comparable constraints to using the vertical form the 3D inversion. Our approach of modeling the sills directly by joint inversion of LOS maps provides a more direct link between data and model. Based on these considerations, we prefer to jointly invert two different InSAR orbits and therefore believe we have enough independent observations to constraint the sill models.

Comment 5: InSAR timeseries: *Did you mask any of the deformation prior to doing the corrections on the interferograms with Pirate? The deformation is fairly broad which you may be removing when solving for the linear ramps etc.*

Response 5: We masked the active rifts of Dabbahu-Manda-Harraro and Central Afar before estimating the parameters of the planar orbital correction and the variance-covariance matrix of the noise. We clarify this point by modifying lines 412-415 as follows: “Orbital ramps were removed by fitting them with a linear function, estimated following an epoch-by-epoch network approach⁵⁸, after masking the active rifts of DMH and CA. A similar network strategy was also used to minimize the topography-correlated atmospheric noise, fitting the linear trend of phase delay with elevation⁵⁹. We applied Atmospheric Phase Screen (APS) filtering using a high-pass Gaussian temporal filter with cut-off window of 1 year, followed by an adaptive low-pass Butterworth spatial filter with cutoff estimated from a sparse Variance-Covariance Matrix (VCM) of the spatially-correlated noise⁵⁷ in the masked LOS velocities (Supplementary Fig. 2)”

Comment 6: *While the timeseries in figure 1 are compelling, could you also plot timeseries from other points to demonstrate that the signal is related to just the inferred inflation. There appear to be other zones with apparent movement, but would be nice to see that these aren't showing the same temporal pattern which may invalidate the conclusion that there is a simultaneous inflation.*

Response 6: As suggested, we extracted time-series from 4 pixels just outside our interpretation uplift signal, but which fall on localized regions of apparent movement in at least one of the two orbits (see new Supplementary Fig. 5). Ascending and descending time-series of points 5-8 show different trends, mainly fluctuating around zero motion and anyway showing trends inconsistent with the temporal pattern of the Central Afar uplift (Fig. 1 in the manuscript).

Comment 7: Also, could you comment on the apparent similarity in timing of uplift? Assuming a common source at the base of the Moho, given the spatial separation of the sills and differences in depth, would you not expect that their surface expression would be offset? Depending on the flow rate it may be that the InSAR data doesn't have high enough temporal resolution but it may be worth adding something in the discussion.

Response 7: Both filtered and raw time-series show a descending trend starting between the end of 2016 and the beginning of 2017 (Supplementary Fig. 2), suggesting the four sills respond to the same episode of magma inflow from an upper mantle source. However, our InSAR data have temporal separation between epochs of about one month during 2016-2017 and we cannot constrain temporal patterns of surface motions at the different sills at less than this time span. If the 4 sills are new intrusions fed from magma flowing from a common mantle source, then flow rates would have to be a minimum of 0.2 m/s for simultaneous motion (within a month) to occur, which is towards the fast end of that expected for basaltic melts (Petrelli and Zellmer, 2021). Otherwise, the sills share a pressure connection between them and a common source, likely in the mantle. In this case, magma channels/pathways from the common source to the sills may have been in place prior to onset of inflation. We favor the interpretation that the multiple sill and deep reservoir system existed a priori and was activated from magma flux into the deeper reservoir. The simultaneous inflation results from the sills sharing a pressure connection with the deeper reservoir and with each other. Such an interpretation is also favored in other examples of simultaneous reservoir motions, for example from Hawaiian volcanoes of Kilauea and Mauna Loa volcanic and from Western Galápagos (Wilding et al., 2023; Reddin et al., 2023). We clarified this in the manuscript at lines 71-73, 153-156 and 201-205

Comment 8: Modelling: It's nice to see the approach to test a range of models but I wonder if the truth is actually a combination of the two. Looking at the residuals in Figure 3, there looks to be a systematic underestimate of the vertical. Could there be a combination of buoyant underplating as well as discrete inflation of the sills? You try using the numerical model to remove the longer wavelength signal before the sill estimates or vice versa. This would likely influence the volumes and depths but would be good to test.

Response 8: Firstly, we re-examined our FEM and we realized that the original solution overestimated the amplitude of the observed surface velocities. Therefore, we calculated a new model and found that a vertical buoyant force generating four topographic peaks at the base of the elastic layer with total elevation change of 24 m over a 4-year period is a better fit to the observations. Nevertheless, this model still does not reproduce the focused uplift of the four sills, as in our original submission (new Supplementary Fig. 18). Hence, our conclusion remains that sill inflation, rather than magma buoyancy, is the dominant process.

Furthermore, we tested whether some magma buoyancy could explain the residual uplift of the sill inflation model. To do this we scaled the buoyancy model to fit the residual signal of the sill inflation model and show that part of the long-wavelength residual uplift could be accounted for by flexure driven by buoyancy (new Supplementary Fig. 19). We found that some buoyancy can explain this long-wavelength residual uplift improving the RMS misfit from 1.35 mm/yr to 1.03 mm/yr (new Supplementary Fig. 18). However, this improvement is minor and within the uncertainty of InSAR. We conclude that the dominant part of the observed uplift is from the sill inflation but that some buoyancy driven uplift may also contribute. We included this part in the method section at lines 509-516. We also showed the modeling result at new Supplementary Figs. 18-19 and explained the possible contribution of buoyancy at line 133-138.

Comment 9: Could you also make some comment on the aspect ratio of the sills? Sills 2 and 4 are quite long and skinny, analogue models often show more disc like bodies rather than relatively narrow sills. This may be related to the regional stress field but may be worth having a sentence or two in the discussion.

Response 9: The elongation of our modeled sills is parallel to the rift and roughly perpendicular to the regional extension, pointing towards a potential relationship between sill geometry and the rifting process. Elongated sills have been observed in exhumed intrusive complexes and large igneous provinces, as observed for example in the Ferrar province (Antarctica) by Muirhead et al. (2012) or in the Karoo Basin, South Africa (Galerne et al., 2011). These field observations combined with analog models have shown that elongated sills with length-to-width ratios up to 11:1 are controlled by magma transport along elongated transient melt channels, focused melt flows, or feeder dikes (Muirhead et al., 2012; Galerne et al., 2011), all of which tend to develop orthogonally to the regional extension direction (e.g., McKenzie et al. 1992; Keir et al., 2011). Similar rift-parallel, elongated sills have also been observed in the upper crust elsewhere in the Afar depression, beneath the Erta Ale Ridge (Pagli et al., 2012). In contrast, symmetric disc-like magma chambers may be a common feature in the shallow crust under developed volcanoes where the radial stress from the topography of the volcano dominates over regional extension. We included part of this explanation at lines 212-214.

Comment 10: Figures: Although the color scales in Figure 1 and 3 are the same, the LOS displacement shown in Figure 3 looks much bigger than in Figure 1. Could you check the color bar used?

Response 10: We have checked all the colorbars and following comments by Reviewer 2, we have now adopted a new and asymmetrical colormap for the modeling figures in this revised version of the manuscript. We also replaced white with yellow in Fig. 1, Fig. 2 and all the supplementary figures related to time-series and 3D velocity maps. This allows for a more detailed view of areas with low velocities.

Reviewer #2

Comment 1: You point to simultaneous uplift as evidence that the four sills are responding to the same variations in deep melt supply, which I find convincing. However, I wonder whether you have investigated the relationship between the displacement time series above the four sills in more detail, to assess how robust the observations of similarity is, and assess when the time series first become correlated? Is the time of uplift onset actually the same in each location or is there a lag? From Figure 1d and e, I think the situation seems more complicated than described at line 70 as “range decrease in 2014, followed by a stable period and a rapid range decrease from 2017 to 2021”. It looks like range decrease is apparent earlier at Alol and Immino in the ascending data? Could you make an assessment of time series similarity using correlation analysis or at least compare the timings of the onset of uplift using time series turning points?

Response 1: We explored the relationship between time-series in more detail, as suggested. In particular, we carried out two different analyses on our descending time-series (Fig. 1e) while the ascending time-series was not used because of a data gap in 2017-2018. As an initial simple test, we segmented the four time-series (Fig. 1e) into 17 windows and performed a linear regression of the data in each window (Figure 1 of this rebuttal). This test shows that the regression lines have variable slopes before 2017, while all the time-series show a similar negative slope starting from the beginning of 2017, indicating a common inflation phase.

Figure 1 - Polynomial fitting on the descending time-series using 17 windows. The lengths of the colored segments show the temporal width of each window. The red area highlights the period of clear common negative trend.

We then did a more robust statistical correlation analysis following Reddin et al. (2023), and calculated the Pearson’s correlation coefficient (r) between pairs of four time-series curves and the statistical significance values (t) (see new Supplementary Fig. 4). The time-series (Fig. 1e of the manuscript) were interpolated to obtain a constant temporal increment of 12 days, which corresponds to the minimum temporal separation between epochs. The Pearson’s correlation coefficient (r) and the statistical significance (t) were then calculated using a period of 150 days (~5 months) to exclude any short-term contributions to the deformation (Reddin et al., 2023). The r value is the correlation between pairs of time-series and ranges from -1 to 1, with -1 for anti-correlated patterns, 0 for uncorrelated patterns, and 1 for correlated patterns. For each r we calculated

the statistical significance test (t). Values of t lower than 0.05 are indicative of r being significant and values > 0.05 indicates r being not significant (Fenton and Neil, 2018). We then identified the statistically significant onset of uplift in our time-series by searching for the first occurrence of $r > 0.95$ in each pair of time-series. The results show that the most significant positive correlation ($r > 0.95$) of the four time-series starts in December 2016 (new Supplementary Fig. 4) and continues until the end of the observation period (2021). No lag in the onset of deformation is observed between the different pairs of time-series indicating that the uplift was simultaneous at the four sills, or that eventual differences in the onset of uplift were shorter than one month (shortest period resolved by our time-series during 2016–2017). Although the ascending time-series could not be analyzed statistically because of a gap in the data, the patterns of the time-series are consistent with that of the descending time-series for the 2017-2021 period (Fig. 1 of the manuscript), confirming our interpretation. Regarding the initial observation period 2014-2015, Immino, Der'Ela Gaggadé and Assal have positive correlations but the correlation ends in 2015 and it is followed by a period of inconsistent correlations at the four sill. Also this pattern is not observed in the ascending time-series (Fig. 1d of the manuscript).

Finally, we also calculated the correlation coefficient between time-series at one of the uplift sites (AlloI) and one of the sites located outside the uplift signal (Supplementary Fig. 4g). This test shows no clear evidence of correlation between the time-series, suggesting that the areas outside the uplift signal are dominated by noise. We conclude that there is a significant correlation between the sills from December 2016 until the end of the observation period in 2021. The cross-correlation analysis is described in a new Supplementary Method section, and new Supplementary Fig. 4. It is referred to in the manuscript at line 71-73.

Comment 2: *How have you identified the four specific sill locations for modelling? The methods do not state the range over which Lat and Lon were allowed to vary – were they then fixed before inversion? More generally, how did you decide that there were sills precisely at sites 1-4? These don't look like the locations of maximum LOS uplift rate From Figure 1, but perhaps maximum vertical velocities from joint inversion with GNSS as show in Figure 2. Is the choice of sill location related to graben axes? Please could this choice be explained in the main text of the paper. The residuals in Figure 3c look systematic, with range increase localised along a few faults and a general underestimation of uplift in the central area – it would be good to include a short discussion of how you interpret that.*

Response 2: Similar to the other parameters, we searched for the best-fit sills location by letting the latitude and longitude of the sill centroids be free to vary within bounds that we set. The bounds on the location of the sill centroids have been let to move ~20-40 km and are therefore quite large: E41.60-41.90, N11.90-12.33 (sill 1); E42.02-42.22, N11.51-11.82 (sill 2); E42.23-42.48, N11.55-11.83 (sill 3); E41.17-41.34, N11.84-12.10 (sill 4). The bounds on the location of the sills have been chosen to be large enough to cover the observed uplift area, and narrow enough to prevent overlap between the sills. Furthermore, the decision of using four sills was based on the observation of four maxima on the vertical velocity maps. We now clarified these points and stated the location bounds in the method at lines 478-482 as suggested.

Regarding the residual in Figure 3c of the manuscript, please see our reply to comment 8 by Reviewer #1, where we explored the contribution of buoyancy to the uplift.

Comment 3: *General point about figures: I don't find the diverging colour scale very helpful for picking out details of a displacement field that shows only range decrease, as only a small fraction of the colour scale is used (Figures 1 and 3 and several in the supplementary, especially Suppl. 9). Please consider using a different colour scale throughout the paper, and certainly for those figures where range change is only in one direction.*

Response 3: We have modified the colorbars of all the figures as suggested. For the modelling we have adopted a magma colorbar with asymmetric scale. For 3D and LOS velocity maps where values range in both directions we instead now use a symmetric colorbar replacing the white with tones of yellow. This allows for a better view of low velocity values.

Comment 4: *Could you put the uplift (or estimated intrusion rates) in the context of Central Afar's recent geodetic history? For example, how to the rates of volume increase in the four sills compare to intrusive processes during the last episode of rifting?*

Response 4: By assuming sill opening rates and geometries of Model 2 and considering an inflation period of ~4 years (2017-2021) we can calculate a total volume change of the four sills of ~0.036 km³. Geodetically estimated volume changes during the 2005 Dabbahu-Manda-Harraro diking episode were ~1.5-2 km³ (Grandin et al., 2009), more than 50 times larger than those estimated in Central Afar during 2017-2021. Before the

Dabbahu-Manda-Harraro events, the Assal-Goubbeth rifting episode of 1978 was estimated to be characterized by a volume change 0.2 km^3 (Tarantola et al., 1979), larger than the Central Afar sills intrusion. We included this comparison in the discussion at lines 180-185.

Comment 5: Line 65. *Please specify number of ifgms/time period instead of 'vast*

Response 5: We have now modified the sentence by including the total number of interferograms as follows: "We processed an InSAR dataset made of 255 interferogram from two orbits, ascending and descending, to obtain the time-series of cumulative satellite Line-of-Sight (LOS) deformation...". All other specific information on the number of interferograms and time period are given in the method section at lines 391-392.

Comment 6: *Please specify what models are shown here – Okada solutions for the four sills only? Please see comments about colour scales.*

Response 6: We clarified this by modifying the sentence in the caption of Fig. 3 as follow: "The blue polygons in b and d are the surface projection of the four Okada sources (sills)".

Comment 7: Line 110, 133-134. *"The depth of the sills follows the trend of the Moho in CA with a progressive southeastward shallowing" – This is generally true, though sill 3 is deeper than sills 2 or 4 – can you comment on this?*

Response 7: The position of the sills might not be accidental but rather influenced by the crustal structure, the presence of barriers (e.g., Kelem and Aharonov, 1998), and the extensional rates in Central Afar. Sills 1 and 3 for example are emplaced in proximity of the crust-mantle transition zone, as typically observed during lower-crustal intrusion, while sills 2 and 4 are located close to the brittle ductile transition zone. These two crustal levels could thus represent preferred boundaries where magma ponds, evolves and potentially feed magma to the shallower crust. We included part of this explanation in the discussion at lines 168-171.

Comment 8: Line 131. *Could you clarify what is meant by network? To me, this implies that the sills are connected to each other directly rather than responding to the same pulse of ascending melt, as described elsewhere.*

Response 8: We used the term network because we observe simultaneous uplift at the four sills and thereby interpret that the sills are sharing a connection. We interpret the connection to more likely be a pressure connection and a link to a common source, likely in the mantle. Please refer to the Response 7 to Reviewer #1 for additional details.

Comment 9: Line 136. *'Locates' > is located? or is currently accumulating?*

Response 9: We replaces "locates" with "is located".

Comment 10: Line 144. *'deepest' refers to just sills 3? Can you point to evidence or the depth of the ductile to brittle transition?*

Response 10: The sentence refers to both sill 1 and 3 and we have clarified this in the manuscript. We now provide more details on the crustal structure of Central Afar by showing the depth distribution of seismicity reported in Pagli et al. (2019) and already used in the present study for Figure 1a. Please refer to *Response 7, 13 and 22* to reviewers #3 and #4.

Comment 11: Line 168. *We have also seen something very similar in the Western Galapagos: <https://www.nature.com/articles/s41467-023-42157-x> I think, if you see fit, that the paragraph at 171-182, could be expanded to address settings beyond rifts, given recent similar observations in Hawai'i and the Galápagos.*

Response 11: These three areas really look like they share similarities in the episodic behavior of magma extraction from depth and migration to shallower sill systems. We already mentioned the Hawai'i case study in our manuscript and we have now included the observations of magmatic connectivity at Galápagos as suggested.

Comment 12: Line 385. *Please clarify at the beginning of the paragraph that 'Models 1 and 2' refer to weighted and unweighted inversions.*

Response 12: We clarified as suggested.

Reviewers #3 and #4

Major concerns

Comment 1: Our (grad student + advisor review) primary concern is that the authors have not clearly stated that potential field models of temporal changes in surface elevation are non-unique, and that the signal decays rapidly with depth to source body. Could we ever detect a signal from a magma body at depths of 20 km or greater? What volumetric changes would be required to detect an inflation at 40 km?

Comment 6: 134-136: The comments about the sill depths mirroring Moho depths seems an over-interpretation, given the non-uniqueness of the sill models. If there were independent MT or seismicity data, we could re-consider.

Comment 21: Supplementary material captions: These need more details and justifications. The SM needs a clear statement about non-uniqueness of models

Response 1, 6 and 21: We are aware that inverse solutions are non-unique and although in our original submission we explored the non-uniqueness of our solutions, we acknowledge that our error calculation could be improved and we completely re-did it. In our original submission we calculated uncertainties for the four sills together using a Monte Carlo simulation of correlated noise, whereby 100 simulations of correlated noise with the same covariance function as in the data are added to the observations and inverted (Wang et al. 2009). Each inversion was done using a full Variance-Covariance Matrix (VCM) and the model parameters errors were then determined from the distribution of the 100 best-fit solutions. We now calculate errors by inverting each sill separately and using a simpler diagonal VCM because it is computationally convenient. We calculated the mean and the standard deviation (σ) of each model parameter from the 100 solutions. In general, we notice some improvement in the distribution of 100 solutions (e.g., sill 1), yet errors remain large and some trade-offs between parameters are also present (new Supplementary Figs. 16 and 17). For example, the depth of the sills trades off with the opening rates. Nevertheless, the mean of the sill depths distribution resulting from the error calculation remain close to our best-fit model and the depth of the sills from the 100 solutions remain located in the mid-to-lower crust: mean depth and σ of 22 ± 11 km for sill 1, 7 ± 4 km for sill2, 11 ± 6 for sill 3, and 7.5 ± 7 km for sill 4 (new Supplementary Figs. 16 and 17). Please note these depths are the mean of 100 solutions hence they represent the statistically significant depth and differ from the best-fit solutions from the inversion of the observations. We thus agree with the reviewers that the best-fit solution shown in the manuscript might be non-unique. We attribute this to the fact that the uplift signal is relatively small compared to the noise level and that the sills are deep. We thus clearly stated this in the manuscript and used independent geophysical data to support our modeling.

In particular, seismic imaging of the crust in Central Afar shows high Vp/Vs of the crust (Hammond et al., 2011, Ahmed et al., 2022). However, velocity models that best locate shallow seismicity in the region have rather normal Vp/Vs (e.g. Belachew et al., 2011), implying that Vp/Vs ratios are particularly high in the lower crust. This is also consistent with S-wave velocity models in Afar which show low Vs at lower crustal depths in Central Afar (Chambers et al., 2022). Magnetic and gravity data collected during 2008-2009 by Demissie et al. (2018) are interpreted to show a lower crust that has been intruded by melt at depths up to ~ 28 km beneath the Dobi graben, just ~10 km south of Immino. Other independent constraints on the magma depth in Assal-Goubbeth are from Doubre et al. (2007) which identified fluid-induced earthquakes in 1996-2001 and inferred the presence of magma at the base of the crust at depth > 8 km. These shallower depths are similar to the range of depths estimated for sills 2-4. All these independent geophysical data on the Central Afar area are consistent with our sill models. We are thus confident that sills at ~20 km depth can be resolved by our data and that a sill could not be at 40 km depth as none of our models are deeper than 28 km. The new errorbar calculations are now shown in Supplementary Figs. 16 and 17. Refer to Response 7, 13 and 22 for our reply regarding the seismicity.

As suggested, we describe in more detail the results of the error calculation at lines 117-124. We now discuss independent geophysical evidence of magma in the lower crust supporting our model and we added the results by Demissie et al. (2018) at lines 157-161. For each sill parameter, we also included mean values and related σ values in the Supplementary Data 7.

Comment 2: Following on from above, why choose as alternative model of an intrusion below the plate, with source depth not stated in the text, and not justified with evidence from exhumed passive margins. Technically speaking, this would be a dynamic source.

Response 2: We clarified that the buoyancy model assumes a 10 km thick elastic layer (previously called elastic plate but now renamed for clarity as the elastic layer) that flexes as buoyant magma ponds at its base. We choose to test this model because recent studies show that the effect of magma buoyancy during intrusion of the crust can be important (e.g., Sigmundsson et al. 2020). Mid and lower crustal sills are widely documented in exhumed passive margins (e.g, Gouly, 2005).

Comment 3: Why 4 sources? Why not 3 or 5? This is key: can the arguments be squeezed into a short paper format?

Response 3: We used 4 sills based on the observation of 4 main patterns consistent with uplift both in the two independent InSAR LOS velocity maps and in the vertical velocity maps from the 3D inversion. We clarify this point at line 458-459. Please refer also to *response 2 to Reviewer #2* for further details on the search bounds of the inversion.

Comment 4: In the elastic half space model domain, the authors decided to use Poissons ratio of “0.25”. Considering that bulk crustal V_p/V_s in the Afar Depression is ≥ 1.8 , Poissons ratio of ≥ 0.28 should be used. Poisson ratio of 0.25 corresponds to V_p/V_s of 1.73 which might be an underestimation of their model’s elastic domain.

Response 4: We acknowledge that variations in the V_p/V_s ratio influence the elastic properties of the half-space. We thus used V_p/V_s ratios varying between 1.75-2.0, as measured in CA by Hammond et al. (2011) and Ahmed et al. (2022), to calculate Poisson’s ratio values of ~ 0.25 -0.33. We then modified the Poisson’s ratio of our half-space and recalculated the model predictions, assuming the source parameters of Model 2 (Fig. 2 of this rebuttal). The figure shows that changing the Poisson ratio from 0.25 to 0.33 is not an important factor and our best-fit model predictions are not significantly changed.

Figure 2 – Comparison of 4 models assuming the same source model (Model2) and progressively increasing Poisson’s ratios from 0.25 to 0.33 (top-right corners). The numbered asterisks show the values of predicted InSAR velocity at the same pixel assuming different Poisson’s ratio.

Comment 5: In their InSAR and GNSS joint inversions, a constant nodal spacing of 3 km was used in their triangular mesh. The GNSS station spacing in the region is much greater than 3 km as shown in Figure SM3. Do the authors think that the jointly inverted 3-D models have spatial resolution of 3 km?

Response 5: The inversion combines dense InSAR velocity maps and sparse GNSS measurements. The nodal spacing is based on the dense InSAR data (90 m pixel spacing) (Wang and Wright, 2012). If ascending and descending InSAR data of the same area are available, as in our case, only a few GNSS measurements are needed to retrieve the 3D velocity field (Wang and Wright, 2012). See also our reply to Reviewer #1 comment 3, the one about the mesh spacing, we show in Figure 1 of this rebuttal.

Comment 7: 144-146; 154-157: Where is the brittle ductile boundary, and given the rapid stressing magmatic zones, would historical seismicity constrain this boundary. I would be more cautious here and focus on key and inarguable points. For example, rapid stressing of visco-elastic material would induce earthquakes, but nowhere do we see evidence of elevated earthquake levels.

Comment 13: Sill intrusions at depth may be seismically silent. This should be stated up front – one of the co-authors is a seismologist.

Comment 22: SM: Needs comments about seismicity: Are data available and do they inform this work? The 6th author is a seismologist, but it's hard to see his impact on the paper.

Response 7 and 13 and 22: Long term seismic recordings from global networks (e.g., ISC and NEIC) have neither sufficient earthquakes nor sufficiently accurate depths to allow constraining of the brittle-ductile layer in Afar. However, we have used seismicity from the temporary network deployed during 2005-2010 (Ebinger et al., 2008; Keir et al., 2009; Belachew et al., 2011) and relocated by Pagli et al. (2019) to explore the depth distribution of earthquakes in the Central Afar area (Lat: N11-N12.8/ Lon: E41-E42.7) (See new Supplementary Fig. 20). The histograms of the earthquake hypocenters show that the 90th percentile of the events reach a depth of ~11 km. Considering that the brittle layer is defined as containing the majority of earthquakes, the top ~10 km in Central Afar can be considered as a good approximation of the brittle layer, with the brittle-ductile transition expected around 10-15 km. This is also broadly consistent with estimates ~5-10 km effective elastic thickness in Afar (Ebinger and Hayward, 1996).

We also analyzed the temporal distribution of earthquakes recorded by ISC during the same time period of InSAR to explore possible increases in the earthquakes rate during the period of the sill intrusions (Figure 9b of this rebuttal). In general seismicity is very sparse and there is no clear evidence of an earthquake increase during 2017. A step-increase in 2019 is caused by earthquakes south of the Dama Ali volcano which is ~ 60 km from the area of the sills. This might point either toward the fact that magma intrusions are seismically silent (as the reviewers proposed in Comment 13) or that the magnitude threshold of the ISC seismic network is poor and low-magnitude earthquakes that might be associated with magma intrusion are not being recorded. We showed this data in the new Supplementary Fig. 20 and included this explanation at lines 175-179.

Comment 8: 169-170: The claim that this is the first paper to document inter-actions between magma chambers in a continental rift is a bit over-stated. Gelai and Oldoinyo Lengai volcanoes in the weakly extended N Tanzania rift occurred in 2005-6, and interactions between sill complexes are documented in Oliva et al. GRL, 2019; and Reiss et al., Frontiers, 2021.

Response 8: We acknowledge that our study builds on previous interpretations of sill complexes in continental rifts. However, the novelty of our study is that our observations of synchronous uplift of 4 lobes covering a large spatial region makes the interpretation of a connected deep sill network more directly rooted to direct observations than the majority of previous studies. In addition, the spatial scale of the observed sill related deformation is significantly larger than most previous studies which are focused beneath specific volcanic complexes. This is important since previous interpretations of large scale lower crustal sill networks beneath magmatic rifts such as Afar were primarily based on passive seismic imaging (e.g. Chambers et al., 2022), but with the significant horizontal smearing of such models bringing into question how expansive the melt systems actually are. We have now clarified in the discussion that our results provide evidence of large-scale simultaneous sills inflation in a continental rift, at lines 219-220.

Minor/editing Comments

Comment 9:24: suggest you replace preferred with common

Response 9: we replaced as suggested.

Comment 10: 35: Readers won't understand 'Afar': The Afar depression encompasses sub-aerial sectors of the Red Sea and Gulf of Aden rifts.

Response 10: We replaced it with "Afar depression" as suggested.

Comment 11: 44: mafic or molten or both? How will they influence Vp/Vs and Poisson's ratio?

Response 11: Both mafic rocks and melt can cause an increase of the Vp/Vs and Poisson's ratios. We acknowledge that the sentence is misleading and we thus modified the sentence as follows: "...consistent with it being continental crust heavily intruded by mafic rocks or melt"

Comment 12: 52: Sills in the lithosphere? What is lithospheric thickness beneath Afar? Why are models indicated as below the lithosphere – confusing.

Response 12: We agree the sentence is confusing as our sills are all above the Moho. We have now modified the sentence by referring to the crust and, in light of the comments above, by better distinguishing between rheological and lithological boundaries.

Comment 14: Fig 1: Text and graph data are very small.

Response 14: We used a larger font as suggested.

Comment 15: Refs 13-16: too much reliance on rates from modelling, and not enough from data. Include key data from Viltres et al., 2020. Viltres, R., Jónsson, S., Ruch, J., Doubre, C., Reilinger, R., Floyd, M., & Ogubazghi, G. (2020). Kinematics and deformation of the southern Red Sea region from GPS observations. *Geophysical Journal International*, 221(3), 2143-2154.

Comment 19: We are not InSAR specialists and cannot contribute to details on lines 340-404. But, we note the reliance on models for rates of rift opening, rather than independent data. GNSS models of Viltres et al. 2020 should be cited.

Response 15 and 19: We clarified that we never relied on GNSS models. Our 3D velocities are obtained using GNSS data and not GNSS models. The GNSS data are from Doubre et al. 2017 and King et al. 2019. We acknowledge Viltres et al. (2020) is a more recent publication yet the new GNSS data in the paper are only in Northern Afar (Danakil block) and are outside the area of interest. The most recent GNSS data from Central Afar data are still from Doubre et al. 2017 and King et al. 2019 and we prefer to cite the original references.

Comment 16: Since the paper is using geophysical constraints to validate their 3-D models while defining the major physical boundaries in their model domain (e.g., Moho and base of plate), I would suggest the authors to use physical boundaries consistently throughout the manuscript. Instead of just saying “mantle” specify whether it is “mantle lithosphere” or “asthenosphere”. Additionally, while the authors attempt to describe the plate in their model domain in terms of the elastic, brittle-ductile and visco-elastic domains, adding terminologies such as upper, middle, or lower crust and upper mantle will help readers easily follow your arguments.

Response 16: We acknowledge that we were not clear in the use of physical boundaries to describe our flexure model, as also in describing its aims. We have now corrected and clarified these aspects at lines 125-130, and as also in the method section, by specifying what these physical layers and boundaries represent.

Comment 17: 103: How can a model ‘find’: What measure of goodness of fit was used to determine preferred models?

Response 17: We replaced the term “modeling” with “inversion”. Our inversion consists of a Monte Carlo simulated annealing algorithm followed by a derivative based procedure (e.g., Cervelli et al 2001). This approach finds the best-fit model by looking for the combination of parameters that minimize the residual RMS misfit between InSAR and observation.

Comment 18: Supplementary Material: Fig 3: The ‘shaded’ area requires a regional expert to spot.

Comment 18: We have now modified the colorbar to provide a better view of areas with low velocities and in shading.

Comment 20: Why would the authors consider a melt influx below the plate and into the asthenosphere where viscosity is only crudely constrained? Why don’t you just show that a sub-crustal influx is undetectable, and consider this an important result: we can’t see deep inflation events beneath rift zones? InSAR can’t ‘see’ to these depths. Isostasy relates to density variations within the plate, not below the plate.

Response 20: We do not mean melt influx under the lithosphere or into the asthenosphere. Our flexure driven by buoyancy model tests the effect of buoyancy below an elastic layer that is 10 km thick. We agree with the reviewers that this was not clear and we have now clarified this section.

References

Ahmed, A. et al. Across and along-strike crustal structure variations of the western Afar margin and adjacent plateau: Insights from receiver functions analysis. *J. African Earth Sci.* 104570 (2022). <https://doi.org/10.1016/j.jafrearsci.2022.104570>.

Belachew, M., C. Ebinger, D. Coté, D. Keir, J. V. Rowland, J. O. S. Hammond, and A. Ayele, Comparison of dike intrusions in an incipient seafloor-spreading segment in Afar, Ethiopia: Seismicity perspectives, *J. Geophys. Res.*, 116, B06405 (2011). <https://doi.org/10.1029/2010JB007908>

Cherkose, B. A. and Saiby, H. Investigation of the Ayrobera geothermal field using 3D magnetotelluric data inversion, Afar depression, NE Ethiopia. *Geothermics*, 94, 102115 (2021)

- Didana, Y. L., S. Thiel, and G. Heinson. Magnetotelluric imaging of upper crustal partial melt at Tendaho graben in Afar, Ethiopia, *Geophys. Res. Lett.*, 41, 3089–3095 (2014). <https://doi.org/10.1002/2014GL060000>.
- Demissie, Z., Mickus, K., Bridges, D., Abdelsalam, M. G. & Atekwana, E. Upper lithospheric structure of the Dobi graben, Afar Depression from magnetics and gravity data. *J. African Earth Sci.* 147, 136–151 (2018).
- Doubre, C. et al. Current deformation in Central Afar and triple junction kinematics deduced from GPS and InSAR measurements. *Geophys. J. Int.* 208, 936–953 (2017).
- Ebinger, C. J., and N. J. Hayward. Soft plates and hot spots: Views from Afar, *J. Geophys. Res.*, 101(B10), 21859–21876 (1996). <https://doi.org/10.1029/96JB02118>.
- Fenton, N. & Neil, M. *Risk Assessment and Decision Analysis with Bayesian Networks*. (Taylor & Francis, 2018). <https://doi.org/10.1201/b21982>.
- Fialko, Y. & Simons, M. Evidence for ongoing inflation of the Socorro Magma Body, New Mexico, from interferometric synthetic aperture radar imaging. *Geophys. Res. Lett.* 28, 3549–3552 (2001).
- Galerne, C., et al. 3D relationships between sills and their feeders: evidence from the Golden Valley Sill Complex (Karoo Basin) and experimental modelling. *Journal of Volcanology and Geothermal Research*, 202 (3-4), 189-199 (2011). <https://doi.org/10.1016/j.jvolgeores.2011.02.006>
- Grandin, R. et al. September 2005 Manda hararo-dabbahu rifting event, Afar (Ethiopia): Constraints provided by geodetic data. *J. Geophys. Res. Solid Earth* 114, (2009).
- Gouly, N.R. Emplacement mechanism of the Great Whin and Midland Valley dolerite sills. *Journal of the Geological Society*, 162 (6), 1047–1056 (2005). <https://doi.org/10.1144/0016-764904-141>
- Hammond, J. O. S. et al. The nature of the crust beneath the Afar triple junction: Evidence from receiver functions. *Geochemistry Geophys. Geosystems* 12, (2011).
- Keir, D., et al. Evidence for focused magmatic accretion at segment centers from lateral dike injections captured beneath the Red Sea rift in Afar. *Geology*, 37 (1), 59–62 (2009) <https://doi.org/10.1130/G25147A.1>
- Keir, D., et al. (2011), The magma-assisted removal of Arabia in Afar: Evidence from dike injection in the Ethiopian rift captured using InSAR and seismicity, *Tectonics*, 30, TC2008. (2011) <https://doi.org/10.1029/2010TC002785>.
- Kelemen, P. B. & Aharonov, E. Periodic formation of magma fractures and generation of layered gabbros in the lower crust beneath oceanic spreading ridges. *Geophys. Monogr. Ser.* 106, 267–289. (1998)
- King, R., Floyd, M., Reilinger, R. & Bendick, R. GPS velocity field (MIT 2019.0) for the East African Rift System. (2019) <https://doi.org/10.1594/IEDA/324785>.
- McKenzie, D., McKenzie, J. M., and R. S. Saunders. Dike emplacement on Venus and on Earth, *J. Geophys. Res.*, 97, 15,977–15,990 (1992). <https://doi.org/10.1029/92JE01559>.
- Muirhead, J. D., Airoidi, G., Rowland, J. V. & White, J. D. L. Interconnected sills and inclined sheet intrusions control shallow magma transport in the Ferrar large igneous province, Antarctica. *Bull. Geol. Soc. Am.* 124, 162–180 (2012).
- Pagli, C., Wright, T., Ebinger, C. et al. Shallow axial magma chamber at the slow-spreading Erta Ale Ridge. *Nature Geosci* 5, 284–288 (2012). <https://doi.org/10.1038/ngeo1414>
- Petrelli, M. and Zellmer, G.F. . Rates and Timescales of Magma Transfer, Storage, Emplacement, and Eruption. In *Dynamic Magma Evolution*, F. Vetere (Ed.), (2020). <https://doi.org/10.1002/9781119521143.ch1>

Reddin, E. et al. Magmatic connectivity among six Galápagos volcanoes revealed by satellite geodesy. *Nat. Commun.* 14, 1–11 (2023).

Sigmundsson, F. et al. Unexpected large eruptions from buoyant magma bodies within viscoelastic crust. *Nat. Commun.* 1–11 (2020) <https://doi.org/10.1038/s41467-020-16054-6>.

Tarantola, A., Ruegg, J. C. & Lepine, J. C. Geodetic evidence for rifting in Afar: A brittle-elastic model of the behavior of the lithosphere. *Earth Planet. Sci. Lett.* 45, 435–444 (1979).

Wang, H. & Wright, T. J. Satellite geodetic imaging reveals internal deformation of western Tibet. *Geophys. Res. Lett.* 39, 1–5 (2012).

Wang, H., Wright, T. J. & Biggs, J. Interseismic slip rate of the northwestern Xianshuihe fault from InSAR data. *Geophys. Res. Lett.* 36, 1–5 (2009).

Wilding, J. D., Zhu, W., Ross, Z. E. & Jackson, J. M. The magmatic web beneath Hawai'i. *Science* (80-.). 379, 462–468 (2023).

REVIEWERS' COMMENTS

Reviewer #2 (Remarks to the Author):

I think the authors have done a nice job of addressing the reviewer comments, especially the quantitative comparison between time series. The figures are also now clearer and easier to understand. I consider that this manuscript is ready for publication.

Reviewer #3 (Remarks to the Author):

It was an honor to critically review the manuscript and help make your manuscript scientifically airtight and ready to be published. Science has won.

Reviewer #4 (Remarks to the Author):

La Rosa, Pagli and colleagues have submitted a carefully revised manuscript that tells an exciting story in the brief Nat Comms format: analyses of geodetic supported by independent geophysical data indicate replenishment of 4 possibly connected sills during the inter-rifting process in the Afar rift zone, Africa. The responses to reviewers was thoughtful and the clarifications address concerns raised in the first review.

I do have a few comments on the revision that I think will increase the impact of the paper, and make it more accessible to other readers.

1. Title and Abstract: Most people I talk to don't know where Afar is. Afar, Africa in title and Afar rift in abstract to establish plate boundary context seem essential.

2. 22-34. The first paragraph is too focused on underplate near the crust-mantle boundary. The petrological references are to sill complexes throughout the crust, as are the references in Discussion to interconnected mid-crustal sills in Iceland, Galapagos, Hawaii (and should be to Natron). The Thybo and Artemieva reference seems oblique to the outcomes - too much has been said about underplate but underplate can't explain fractionation patterns or increasingly more detailed images of mid-crustal sill complexes. Sills are the only way ocean islands can be constructed, for example. I would add the time element to show that multiple segments can be activated and indicate inter-connectivity. See also Oliva et al. 2018 who determine the repeat times of magma/gas recharge of stacked sills in a rift zone using calibrated repeating earthquakes and independent seismic data.

3. Fig 1 and text. The East African rift system is also shown in Fig 1 and is the 3rd arm of the triple junction shown in Fig. 1. EA fits and can be explained in the caption. Confusing without it.
4. 81 - isn't proximity important?
5. Fig 2 - Label the sill complexes 1-4 in all panels. Will really help the reader follow the modelling.
6. Uh-oh : Authors quote V_p/V_s of 1.9-2.0 yet the models use Poisson's ratio of 0.25. Watch consistency.
7. Fig 3 is very similar to sills in Oliva et al. GRL 2019 and Reiss et al. 2021.
8. 212 - Look beyond authors' experience in Iceland and Afar - 'deeper parts' and resolution are imaged in the Natron area : Reiss et al. 2022 + Roecker et al. GJI 2017 provide more detailed images of a magma-rich rift in repose from RF, joint AN, arrival time and gravity inversions, attenuation, etc. Oliva and Reiss are early career, too.

Oliva, S.J., C. J. Ebinger, C. Wauthier, J. Muirhead, S. Roecker, E. Rivalta, S. Heimann (2018), Insights into fault-magma interactions in an early-stage continental rift from source mechanisms and correlated volcano-tectonic earthquakes, *Geophys. Res. Letts.*, doi: 10.1029/2018GL080866.

Reiss, M. C., De Siena, L., & Muirhead, J. D. (2022). The interconnected magmatic plumbing system of the Natron Rift. *Geophysical Research Letters*, 49(15), e2022GL098922.

Roecker, S., Ebinger, C., Tiberi, C., Mulibo, G., Ferdinand-Wambura, R., Mtelela, K., Kianji, G., Muzuka, A., Gautier, S., Albaric, J. and Peyrat, S. (2017). Subsurface images of the Eastern Rift, Africa, from the joint inversion of body waves, surface waves and gravity: investigating the role of fluids in early-stage continental rifting. *Geophysical Journal International*, 210(2), pp.931-950.

C J Ebinger

Reviewer #4.

Comment 1. Title and Abstract: Most people I talk to don't know where Afar is. Afar, Africa in title and Afar rift in abstract to establish plate boundary context seem essential.

Response 1. We modified the title and included these details in the abstract, as suggested.

Comment 2. 22-34. The first paragraph is too focused on underplate near the crust-mantle boundary. The petrological references are to sill complexes throughout the crust, as are the references in Discussion to interconnected mid-crustal sills in Iceland, Galapagos, Hawaii (and should be to Natron). The Thybo and Artemieva reference seems oblique to the outcomes - too much has been said about underplate but underplate can't explain fractionation patterns or increasingly more detailed images of mid-crustal sill complexes. Sills are the only way ocean islands can be constructed, for example. I would add the time element to show that multiple segments can be activated and indicate inter-connectivity. See also Oliva et al. 2018 who determine the repeat times of magma/gas recharge of stacked sills in a rift zone using calibrated repeating earthquakes and independent seismic data.

Comment 7. Fig 3 is very similar to sills in Oliva et al. GRL 2019 and Reiss et al. 2021.

Comment 8. 212 - Look beyond authors' experience in Iceland and Afar - 'deeper parts' and resolution are imaged in the Natron area: Reiss et al. 2022 + Roecker et al. GJI 2017 provide more detailed images of a magma-rich rift in repose from RF, joint AN, arrival time and gravity inversions, attenuation, etc. Oliva and Reiss are early career, too.

Response 2,7,8: We only mentioned underplating in the first paragraph in the context that the term has been replaced in recent literature by the term "lower crustal sills". We therefore prefer to leave this as it clarifies how our work relates to literature prior to the last few years. We agree that our interconnected sills are similar to what observed seismically at the volcanic plumbing systems of Natron rift (East Africa) by Oliva et al. (2018) and Reiss et al. (2022) and we have now added the temporal element and these new references of lower crustal sill intrusions in rift contexts. In particular, we included the above reference in the introduction at lines 35-36, and in the discussion at lines 207-209, as suggested by the reviewer.

Comment 3. Fig 1 and text. The East African rift system is also shown in Fig 1 and is the 3rd arm of the triple junction shown in Fig. 1. EA fits and can be explained in the caption. Confusing without it.

Response 3. We clarified this aspect by including the Main Ethiopian Rift (MER) branch in the inset of Fig. 1a. We also described the triple-junction between the MER, Gulf of Aden and Red Sea rift branches in the text at lines 37-39, as suggested.

Comment 4. 81 - isn't proximity important?

Response 4. We clarified the proximity of the sills by describing the width and length of the area showing uplift in Central Afar (see track changes at line 86)

Comment 5. Fig 2 - Label the sill complexes 1-4 in all panels. Will really help the reader follow the modelling.

Response 5. We prefer to avoid to mix between results from the source modelling and from the 3D velocity field inversion shown in Fig.2 and therefore we prefer to leave Fig. 2 unchanged.

Comment 6. Uh-oh: Authors quote V_p/V_s of 1.9-2.0 yet the models use Poisson's ratio of 0.25. Watch consistency.

Response 6. V_p/V_s ratios in Central Afar range between 1.8-2.2 as shown in Fig. 3. For our analytical modelling, we assumed a semi-infinite elastic half-space with a Poisson's ratio of 0.25 and this is a reasonable value for the entire half-space considering that relatively high V_p/V_s ratios, between 1.9-2.0, are localized to only a few areas in Central Afar. Nevertheless, we already addressed this comment in the previous rebuttal where we tested a series of model simulations using Poisson's ratio values ranging ~0.25-0.33 based on the V_p/V_s ratios measured in Central Afar by Hammond et al. (2011) and Ahmed et al. (2022) and shown in Fig. 3. The simulations showed that changing the Poisson ratio from 0.25 to 0.33 is not an important factor and the best-fit model predictions do not change significantly.